# Effect of Miscellaneous Meal Replacements for Soybean Meal on Growth Performance, Serum Biochemical Parameters, and Gut Microbiota of 50–75 kg Growing Pigs

**DOI:** 10.3390/ani13223499

**Published:** 2023-11-13

**Authors:** Zhentao He, Xianliang Zhan, Shuting Cao, Xiaolu Wen, Lei Hou, Shuai Liu, Huayu Zheng, Kaiguo Gao, Xuefen Yang, Zongyong Jiang, Li Wang

**Affiliations:** State Key Laboratory of Livestock and Poultry Breeding, Key Laboratory of Animal Nutrition and Feed Science in South China, Ministry of Agriculture and Rural Affairs, Guangdong Provincial Key Laboratory of Animal Breeding and Nutrition, Maoming Branch, Guangdong Laboratory for Lingnan Modern Agriculture, Institute of Animal Science, Guangdong Academy of Agricultural Sciences, Guangzhou 510640, China; kuzma022133@gmail.com (Z.H.); zhanxianliang1996@163.com (X.Z.); caoshuting@gdaas.cn (S.C.); wenxiaolu@gdaas.cn (X.W.); rhoulei@126.com (L.H.); 15027016245@163.com (S.L.); 17872305992@163.com (H.Z.); gaokaiguo312@126.com (K.G.); yangxuefen@gdaas.cn (X.Y.); jiangzy@gdaas.cn (Z.J.)

**Keywords:** soybean meal, miscellaneous meal, growing pigs, growth performance, gut microbiota

## Abstract

**Simple Summary:**

Given the escalating production of livestock and poultry coupled with the surging cost of soybean meal, the quest for alternative raw materials capable of replacing soybean meal is gaining increasing significance. Rapeseed meal, cottonseed meal, and sunflower seed meal, which are common by-products of agricultural production, offer viable plant-based protein substitutes for soybean meal in pig production. This study reveals that the incorporation of miscellaneous meal (rapeseed meal, cottonseed meal, and sunflower seed meal) as a substitute for soybean meal in the diet did not significantly influence the growth performance, apparent nutrient digestibility, serum amino acid content, or fecal microbiota diversity of growing pigs weighing between 50 and 75 kg. These results suggest that miscellaneous meal (rapeseed meal, cottonseed meal, and sunflower seed meal) could potentially serve as a partial or complete replacement for soybean meal in the diet of growing pigs.

**Abstract:**

This study was carried out to investigate the effects of miscellaneous meal (rapeseed meal, cottonseed meal, and sunflower seed meal) as a replacement for soybean meal on growth performance, apparent nutrient digestibility, serum biochemical parameters, serum free amino acid contents, and gut microbiota of 50–75 kg growing pigs. A total of 54 healthy growing pigs (Duroc × Landrace × Yorkshire) with initial body weights (BWs) of 50.64 ± 2.09 kg were randomly divided into three treatment groups, which included the corn–soybean meal group (CON), corn–soybean–miscellaneous meal group (CSM), and corn–miscellaneous meal group (CM). Each treatment included six replicates with three pigs in each replicate. Dietary protein levels were maintained at 15% in all three treatment groups. Additional rapeseed meals, cottonseed meals, and sunflower seed meals were added to the CSM group’s meals to partially replace the 10.99% soybean meal in the CON group in a 1:1:1 ratio. Pigs in the CM group were fed a diet with a mixture of miscellaneous meals (7.69% rapeseed meal, 7.69% cottonseed meal, and 7.68% sunflower seed meal) to totally replace soybean meal. Our findings revealed that there was no significant impact of replacing soybean meal with miscellaneous meal on the ADG (average daily gain), ADFI (average daily feed intake), or F/G (feed-to-gain ratio) (*p* > 0.05) of growing pigs weighing 50–75 kg, nor on the crude protein, crude fat, or gross energy (*p* > 0.05) of the diet. On the other hand, compared to the CON group, the CM group exhibited significantly elevated serum alanine aminotransferase (ALT) and triglyceride (TG) levels (*p* < 0.05), while urea levels were significantly reduced (*p* < 0.05). No significant effect was observed on the serum free amino acid contents (*p* > 0.05) following the substitution of soybean meal with miscellaneous meal. A *t*-test analysis indicated that compared with the CON group, the CM group exhibited a significantly diminished abundance of *Euryachaeota* at the phylum level and augmented abundance of *Desulfobacterota* at the genus level. This study demonstrated that the miscellaneous meals (rapeseed meal, cottonseed meal, and sunflower seed meal) as a substitute for soybean meal in the diet had no significant negative effects on the growth performance, apparent nutrient digestibility, serum amino acid content, or diversity of fecal microbiota in 50–75 kg growing pigs. These results can be helpful in developing further miscellaneous meals (rapeseed meal, cottonseed meal, and sunflower seed meal) as functional alternative feed ingredients to soybean meal in pig diets.

## 1. Introduction

Soybean meal (SBM) is the most commonly used feed ingredient as a protein source in non-ruminant rations because of its relatively high protein content (44–49%) and value for providing an animal’s body an adequate supply of amino acids [1,2]. In addition, soybean meal is the main protein source in swine feeds in most parts of the world, but there is growing concern about the pig industry’s dependence on importing large quantities of soybean meal. So, it is imperative to find sustainable and workable ways to generate and supply nutritional protein and then maintain appropriate yield and animal performance levels to stay competitive in the global marketplace. Given the continuous escalation of livestock and poultry production, identifying alternative raw materials that could potentially substitute soybean meal has become increasingly critical.

Alternative native protein feedstocks are required to increase pig production’s sustainability and self-sufficiency. Rapeseed meal, cottonseed meal, and sunflower seed meal are common by-products of agricultural production in China. These by-products have high protein contents, high yields, low prices, etc., and are a class of plant-based protein raw materials that can replace soybean meal in pig production. Prior research indicates that the judicious and strategic utilization of such protein raw materials (including rapeseed meal, cottonseed meal, and sunflower seed meal) can effectively diminish the quantity of soybean meal in balanced feed. This carries substantial implications for mitigating reliance on soybean meal [3,4].

In swine production, there are differences in the use of rapeseed meal, cottonseed meal, and sunflower seed meal in the different growing phases. It has been found that the addition of rapeseed meal and fava beans to the diets of growing pig (108.7 ± 4.2 kg final body weight) can improve the feed conversion rate during the fattening period and increase the free amino acids in the blood [5]. The crude protein, mineral, and vitamin content of cottonseed meal is similar to that of soybean meal. It was found that the partial replacement of 2.5% soybean meal in a diet of cottonseed meal had no significant effect on the growth performance of growing pigs (13.18 to 39.81 kg) [4]. Kim et al. [6] found that growing pigs (initial body weight of 19.3 ± 1.8 kg) have a better digestive utilization efficiency of high-protein sunflower seed meal. All of the above studies show that miscellaneous meals (rapeseed meal, cottonseed meal, and sunflower seed meal) are a promising protein material for swine.

However, it remains unclear whether the replacement of soybean meal with a mixture of rapeseed meal, cottonseed meal, and sunflower meal in the diets of growing swine is associated with changes in growth performance and intestinal microbial diversity in growing pigs. Therefore, the purpose of this study was to investigate the effects of using miscellaneous meal (rapeseed meal, cottonseed meal, and sunflower seed meal) in place of soybean meal in diets on growth performance, apparent nutrient digestibility, serum biochemical parameters, serum free amino acid contents, and fecal microbiota in 50–75 kg growing pigs.

## 2. Materials and Methods

Animal protocols in the present study were performed following the Guidelines for the Care and Use of Animals for Research and Teaching, with approval from the Animal Care and Use Committee of Guangdong Academy of Agricultural Science (authorization number GAIASIAS-2022-022).

### 2.1. Experimental Design, Diets, and Management

In the process of experimental grouping, individual body weights of the growing pigs were meticulously recorded using a precision floor scale. This was performed prior to the commencement of the study to acquire initial body weight data for experimental reference. A total of 54 healthy growing pigs (Duroc × Landrace × Yorkshire), with an initial body weight (BW) of 50.64 ± 2.09 kg, were randomly allocated into three treatment groups: the corn–soybean meal group (CON), the corn–soybean–miscellaneous meal group (CSM), and the corn–miscellaneous meal group (CM). Each treatment encompassed 6 replicates with 3 pigs per pen (9 barrows and 9 gilts). The growing pigs utilized in this study were sourced from the breeding test base of the Institute of Animal Science, Guangdong Academy of Agricultural Sciences. The pigs in the CON group were fed a corn–soybean meal. Dietary protein levels were maintained at 15 ± 0.3% in all three treatment groups. Additional rapeseed meals, cottonseed meals, and sunflower seed meals were added to the CSM group’s meals to partially replace the 10.99% soybean meal in the CON group in a 1:1:1 ratio. Pigs in the CM group were fed a diet of miscellaneous meals (7.69% rapeseed meal, 7.69% cottonseed meal, and 7.68% sunflower seed meal) to completely replace soybean meal. The dietary treatments included a control diet based on corn–soybean meal, which contained all nutrients in levels recommended by the nutrient requirements of growing pigs [7]. The composition of basal diets is shown in Table 1, and nutrient information for rapeseed meal, cottonseed meal, and sunflower seed meal are presented in Table 2.

The experiment lasted for 24 days from beginning to end. The pig house was kept at a constant temperature of 26–28 °C. Each pen location maintained air circulation. Feed was meticulously dispensed into the troughs of the experimental pigs thrice daily, at 7:00, 12:00, and 17:00. We meticulously calculated the quantity of feed to be added at each feeding interval. This ensured a consistent supply of feed in the troughs, allowing for ad libitum feeding, and guaranteed that residual feed remained until the subsequent feeding session, thereby supporting the growth requirements of the pigs. Any additions, losses, residual feed, or moldy substances extracted from the troughs were diligently documented for each replicate pen on a daily basis. All 18 pens were identical, with the same covered area (3 m^2^/pig), and were equipped with similar troughs for feed concentrates and water. Pigs were provided ad libitum access to water and feed during the entire experimental trial. 

### 2.2. Sample Collection

After 24 days of the experimental period, the pigs in each group were weighed. Three days before the end of the experiment, fresh fecal samples were collected from pigs before feeding in the morning for 3 consecutive days. Fresh, uncontaminated fecal samples were collected once a day at 7:00 a.m., weighed and mixed with 10% hydrochloric acid 10 mL per 100 g sample for nitrogen fixation treatment, and stored at −20 °C for subsequent analysis. In addition, fresh unground fecal samples were received and stored at −80 °C for gut microbial composition and diversity. One pig with an average body weight for each pen was selected for blood collection at the end of the experiment. After fasting for 12 h, the blood (20 mL) was drawn from a pig’s ear vein and centrifuged at 3000 rpm for 10 min to obtain serum samples. Serum samples were received and stored at −80 °C for the determination of serum biochemical parameters and free amino acid contents. During the index analysis, the serum was placed in an ice water bath to preserve its activity.

### 2.3. Growth Performance

Feed consumption of each replicate was recorded from the beginning to the end of the experiment. The average daily gain (ADG), average daily feed intake (ADFI), and feed-to-gain ratio (F/G) were calculated accordingly.

### 2.4. Apparent Nutrient Digestibility

All diets were given a starting dose of titanium dioxide (0.4%), which was used as an indigestible marker of apparent nutrient digestibility. In pens, feces was collected, dried, sampled, and stored at −20 °C for analysis. All of the growing pigs’ excrement was defrosted, blended, and then heated at 65 °C for 72 h before the natural moisture was restored at room temperature for 24 h.

The crude protein content was estimated by multiplying the total nitrogen content, measured with a Kjeltec 8400 analyzer (FOSS Analytical AB, Hoganas, Sweden), by a factor of 6.25; the crude fat content was determined using an automated extraction analyzer (XT 15i, Ankom Technology, Macedon, NY, USA). The total energy content of the rations and manure samples was determined using an oxygen bomb calorimeter (6400, Parr Instrument, Moline, IL, USA) according to the international standard ISO 9831:1998 [8] method. The concentration of TiO_2_ in all samples (feed and fecal) was measured according to Myers et al. [9].

Apparent nutrient digestibility was calculated as follows [10]: Apparent nutrient digestibility (%) = [1 − (TiO_2_ content in the diet/TiO_2_ content in the fecal sample) × (nutrient content in the fecal sample/nutrient content in the diet)] × 100.

### 2.5. Serum Biochemical Parameters

The serum biochemical parameters were measured using a VITAL automatic biochemical analyzer (ELITechGroup SAS, Puteaux, France). And all of them were measured using reagent kits from BioSino Bio-Technology & Science Inc. (Beijing, China), including those for total protein (TP, Item No. 180531), creatinine (CRE, Item No. 100020170), aspartate aminotransferase (AST, Item No. 100020010), alkaline phosphatase (ALP, Item No. 100020020), alanine aminotransferase (ALT, Item No. 100020000), albumin (ALB, Item No. 180641), urea (UREA, Item No. 100000280), glucose (GLU, Item No. 200891), triglyceride (TG, Item No. 198021), cholesterol (CHO, Item No. 192061), high-density lipoprotein cholesterol (HDL-C, Item No. 100020235), and low-density lipoprotein cholesterol (LDL-C, Item No. 100020245). 

### 2.6. Free Amino Acid Contents

The serum concentrations of free amino acid contents, including essential amino acids, non-essential amino acids, non-protein amino acids, and derivatives/metabolites of amino acids, were measured. Essential amino acids included lysine (Lys), methionine (Met), threonine (Thr), valine (Val), isoleucine (Ile), leucine (Leu), phenylalanine (Phe), tryptophan (Trp), histidine (His), and arginine (Arg). Non-essential amino acids included serine (Ser), alanine (Ala), glycine (Gly), tyrosine (Tyr), glutamic acid (Glu), aspartic acid (Asp), cystine (Cys), hydroxyproline (Hypro), and proline (Pro). The derivatives/metabolites of amino acids included taurine (Tau), NH_3_, citrulline (Cit), and ornithine (Orn).

The protein in each sample was precipitated by adding 0.4 mL of serum sample to 1.2 mL of 10% sodium sulfosalicylic aqueous solution (*w*/*v*), shaking well, and centrifuging at 12,000 rpm/min for 15 min. A 1 mL aliquot of the supernatant was filtered over a 0.22 μm membrane into an L-8900 amino acid auto-analyzer (Hitachi, Ltd., Beijing, China), and the amino acid concentration was determined by using the principle of post-column derivatization of ninhydrin with an external standard.

### 2.7. Analysis of Gut Microbial Composition and Diversity

In this experiment, we used 16S rRNA sequencing technology to examine the impact of switching from soybean meal to mixed miscellaneous meal (rapeseed meal, cottonseed meal, and sunflower seed meal) on the fecal flora of pigs weighing 50 to 75 kg. Microbial DNA was extracted from fecal contents using a DNA kit (Omega Bio-tek, Norcross, GA, USA) according to the manufacturer’s instructions. Qualified extracted DNA samples were then diluted to 1 ng/μL with sterile water and amplified with specific primers (341 F: 5′-CCTAYGGGRBGCASCAG-3′; 806 R: 5′-GGACTACNNGGGTATCTAAT-3′) to amplify the V3-V4 variable region of the 16S rRNA gene. PCR amplicons were purified using the Qiagen Gel Extraction Kit (Qiagen, Hilden, Germany) according to the manufacturer’s instructions. PCR products meeting the library construction requirements were assembled using TruSeq DNA PCR ⁃Free Sample Preparation Library Construction Kit (Illumina, San Diego, CA, USA) and then sequenced using the HiSeq 2500 PE 250 platform (Novohozhiyuan Bioinformatics Co., Ltd., Tianjin, China). Microbial community analysis of the 16S rRNA sequencing data was performed using the QIIME software package (Version 1.9.1). After the removal of chimeric sequences using UCHIME software (Version 4.2.40) (Tiburon, CA, USA), high-quality sequences from operational taxonomic units (OTUs) with 97% identity were aligned with the SILVA database (Ribocon GmbH, Bremen, Germany). The Venn diagram with shared and unique OTUs was used to identify the similarities and differences among treatments.

The alpha diversity parameters included the Observed_species index, Shannon index, Simpson index, Chao1 index, Ace index, and PD_whole_tree index. Principal component analysis (PCA), principal coordinate analysis (PCoA), and non-metric multidimensional scaling (NMDS) were performed to calculate the β diversity between groups. The differences in the relative abundances of microbiota among treatments were compared using the unweighted pair-group method with arithmetic means (UPGMA), *t*-test, and the linear discriminant analysis effect size (LEfSe).

### 2.8. Statistical Analysis

The data were subjected to one-way ANOVA, analysis of covariance (ANCOVA), two-way ANOVA, and *t*-test using SPSS 23.0 (SPSS, Inc., Chicago, IL, USA). Significant differences between means were compared using Duncan’s multiple comparison test. The replicates (*n* = 6) were considered experimental units. The results were expressed as mean and combined softard error (SEM). *p* < 0.05 was considered significantly different, while *p* < 0.10 indicated a trend.

## 3. Result

### 3.1. Growth Performance

As shown in Table 3, partial or total substitution of soybean meal with a miscellaneous meal (rapeseed meal, cottonseed meal, and sunflower seed meal) in the diet did not significantly influence the average daily gain (ADG), average daily feed intake (ADFI), or feed-to-gain ratio (F/G) of growing pigs weighing between 50 and 75 kg (*p* > 0.05). In order to minimize the impact of initial body weight on the growth performance of the growing pigs, we utilized an analysis of covariance, incorporating the initial body weight as a covariate. The subsequent analysis revealed no statistically significant differences in the final body weight (F = 0.001, *p* = 0.975) or average daily gain (F = 0.043, *p* = 0.839) after adjustments were made for the initial body weight (Table 4). Given the potential influence of sex on the growth performance of pigs, we conducted a two-factor ANOVA analysis based on the treatment × sex model (Table 5). The results indicated that sex did not significantly affect the FBW or ADG of growing pigs in the 50–75 kg weight range (*p* > 0.05). Moreover, there was no interaction effect observed between treatment × sex on the FBW or ADG (*p* > 0.05).

### 3.2. Apparent Nutrient Digestibility

Similarly, no significant effect was observed on the apparent digestibility of nutrients, including crude protein, crude fat, and gross energy, when soybean meal was partially or totally replaced with a miscellaneous meal (rapeseed meal, cottonseed meal, and sunflower seed meal) in the diet (*p* > 0.05) (Table 6). 

### 3.3. Serum Biochemical Parameters

The results showed that compared with the CON group, serum ALT and TG levels were significantly higher in the CM group (*p* < 0.05), and urea levels were significantly lower (*p* < 0.05). However, there was no significant effect of miscellaneous meal (rapeseed meal, cottonseed meal, and sunflower seed meal) replacing soybean meal in the diet on the serum TP, CRE, AST, ALP, ALB, GLU, CHO, HDL-C, or LDL-C levels (*p* > 0.05) (Table 7).

### 3.4. Free Amino Acid Contents

Based on the findings in Table 8, our research indicates that partially or totally substituting miscellaneous meals (rapeseed meal, cottonseed meal, and sunflower seed meal) with soybean meal in the diet did not have a significant effect on the contents of serum free amino acids, including Lys, Met, The, Val, Ile, Leu, Phe, Trp, His, Arg, Ser, Ala, Gly, Tyr, Glu, Asp, Cys, Hypro, Pro, Tau, NH_3_, Cit, and Orn. (*p* > 0.05).

### 3.5. Gut Microbiota Composition and Diversity

A total of 1278 common OTUs were identified across the three groups. The CON group had 285, the CSM group had 261, and the CM group had 226 (Figure 1A). At the phylum level (Figure 1B), the dominant bacteria were *Firmicutes*, and the abundances in the three groups were 80%, 77%, and 77%, respectively. At the class level (Figure 1C), the dominant bacteria were *Bacilli*, *Clostridia*, and *Proteobacteria*. At the order level (Figure 1D), the dominant bacteria were *Lactobacillales*, *Bacteroidales*, and *Clostridiales*. At the family level (Figure 1E), the dominant bacteria were *Streptococcaceae*, *Lactobacillaceae*, and *Clostridiaceae*. At the genus level (Figure 1F), the dominant bacteria were *Streptococcus*, *Lactobacillus*, and *Clostridium_sensu_stricto_1*.

As depicted in Figure 2, the replacement of soybean meal with a miscellaneous meal (rapeseed meal, cottonseed meal, and sunflower seed meal) in the diet did not significantly impact the alpha diversity of fecal microbiota, as indicated by the Observed_species index, Shannon index, Simpson index, Chao1 index, Ace index, and PD_whole_tree index.

In terms of beta diversity, the distributional distances between the three groups (CON, CSM, and CM) were not distinctly separated, as evidenced by the results of the NMDS, PCA, and PCoA analyses (Figure 3A). Furthermore, the UPGMA clustering tree, based on the Bray–Curtis distance (Figure 3B), did not reveal any significant differences in fecal microbial composition among the three groups. The *t*-test analysis showed that compared with the CON group, the CM group exhibited a significantly reduced abundance of *Euryachaeota* at the phylum level and an increased abundance of *Desulfobacterota* at the genus level. LEfSe analysis showed that the diet of the CON group enriched the abundance of *Lachnospiraceae_NK4A136_group*.

## 4. Discussion

Feed costs have a significant impact on the profitability of pork farms and are a major factor in the cost of raising pigs. The most popular protein source in the world is soybean meal (SBM), however, increasing the amount of SBM in pig diets has increased the cost of feed [11,12]. Rapeseed meal is a by-product of oil extraction from rapeseed. Rapeseed is mainly grown in China, India, Canada, Europe, and Australia. Rapeseed meal is an important alternative to soybean meal as a feed protein ingredient and is easily digested and utilized by animals. Rapeseed meal contains a complete range of amino acids [13]. Cottonseed meal is a by-product obtained from cottonseed after crushing and leaching to extract oil. Cottonseed is mainly grown in China, India, the United States, Pakistan, and Brazil. Sunflower seed meal is obtained from by-products of producing sunflower seed oil from partially hulled sunflower seeds by pre-pressure leaching or direct solvent leaching. Sunflower seeds are mainly grown in China, Russia, Argentina, the United States, and Ukraine. Rapeseed meal, cottonseed meal, and sunflower seed meal have traditionally been explored as substitutes for soybean meal in livestock and poultry feed, as documented in previous studies [14,15,16]. However, there is a paucity of research investigating the impact of rapeseed meal, cotton meal, and sunflower meal as dietary replacements for soybean meal on growth performance, apparent nutrient digestibility, serum biochemical parameters, serum free amino acid content, and intestinal microbiota in 50–75 kg growing pigs. Consequently, our study seeks to elucidate the effects of diets wherein soybean meal is replaced with rapeseed meal, cotton meal, and sunflower meal in 50–75 kg growing pigs.

The present results showed that the partial or complete replacement of soybean meal with miscellaneous meal does not affect the growth performance or development rate of 50–75 kg growing pigs. This observation aligns with the report by Xie et al. [17], who found that replacing 11% soybean meal in control diets with 13% double-low rapeseed meal did not adversely affect the growth performance of growing pigs (62 kg). Similarly, studies have indicated that the partial substitution of 5% soybean meal with rapeseed meal and cottonseed meal in the diet does not significantly influence the growth performance of growing pigs [4]. Moreover, research has shown that sunflower seed meal does not have a significant impact on the growth performance of growing pigs (62 kg) when it replaces soybean meal in the diet at 5%, 10%, and 15% [18]. Another study showed that there were no differences in growing–finishing gilt BW changes during gestation, in growing–finishing gilt BW on day 1 post-farrowing, or at weaning due to dietary treatments, and there were no differences in ADFI between gestation and lactation diets [19]. And Choi et al. [20] also found no differences in body weight or daily weight gain among growing–finishing pigs (29.94 ± 0.06 kg) fed diets containing rapeseed meal. The aforementioned correlation results are in line with the growth performance outcomes that were observed in our experiments. But from week 13 of the growing season onward, increasing dietary rapeseed additions lowered body weight and weight increase [20]; this could be attributed to the elevated levels of rapeseed meal, which led to an increase in the fiber content of the feed [11]. The inclusion of rapeseed in the diets of monogastric animals has been somewhat restricted due to its substantial fiber and oligosaccharide content, which stands at about 2.5% [21]. Consistent with the literature, a significant study by Hansen et al. [22] revealed a marked decrease in both average daily gain and feed conversion ratios, indicating a potential negative impact on the feed intake of growing–finishing pigs. This effect was observed when the animals were fed nutritionally balanced diets that contained escalating levels of rapeseed meal. Additionally, over the course of the study, G/F ratios tended to fall as dietary rapeseed addition increased. It has been found that the gradual addition of rapeseed meal to feed in the range of 0–30% will linearly reduce the ADFI of growing pigs, and there is a tendency to linearly reduce ADG [23]. And Hong et al. [24] also found that ADG overall responded quadratically to the rising dietary level of rapeseed meal, increasing by 17% when the dietary level of rapeseed meal was increased from 0% to 20% and decreasing by 16% when the dietary level of rapeseed meal was increased from 20% to 40%. The discrepancy between these results might be attributed to the varying proportions of soybean meal, rapeseed meal, cottonseed meal, and sunflower meal in the pig feed, as well as the differing contents of anti-nutritional factors in these meals.

While the growth performance did not exhibit any significant differences, the impact of the miscellaneous meal treatment on nutrient digestion in growing pigs weighing between 50 to 75 kg remains undetermined. Apparent nutrient digestibility serves as an indicator of an animal’s capacity to digest and absorb the nutrients present in their feed. A stronger absorption ability is typically associated with more favorable animal growth. Shim et al. [4] found that the replacement of rapeseed meal and cottonseed meal with soybean meal in different proportions at different stages did not significantly alter apparent nutrient digestibility of crude protein, crude fat, or gross energy in growing pigs, and this is consistent with the results of our experiments. Similarly, it has been found that the replacement of soybean meal with rapeseed meal (380 g/kg replacement rate during the growing period and 720 g/kg replacement rate during the finishing period) in the ration resulted in a decrease in apparent crude protein digestibility but had no effect on apparent crude fat digestibility in finishing pigs [25]. However, the substitution of soybean meal with different varieties of sunflower seed meal revealed varying effects on the apparent digestibility of gross energy in growing pigs [3]. This discrepancy may be attributed to the fact that different processing conditions, such as temperature, pressure, or duration, can alter the chemical composition and consequently the energy content of sunflower seed meal [26]. These findings suggest that the substitution of soybean meal with rapeseed meal, cottonseed meal, and sunflower seed meal does not significantly affect growth performance or apparent nutrient digestibility in growing pigs. 

Upon establishing that the miscellaneous meal did not influence growth performance or apparent nutrient digestibility in growing pigs weighing 50–75 kg, our research shifted focus to the physiological and organismal effects of the miscellaneous meal. Serum biochemical parameters, which are commonly employed to gauge the physiological and health status of animals, were examined. The concentrations of serum ALT indirectly indicate the health status of the liver, and abnormal ALT activities imply potential hepatic tissue damage [27]. The results of this study showed that the complete replacement of a soybean meal with a miscellaneous meal resulted in an increase in serum ALT levels in growing pigs. This is consistent with the results of previous experiments, which have shown that as the level of cottonseed meal added to the ration increases, the level of cotton phenol in the ration rises, and the levels of ATL significantly increase [28,29]. Urea is produced by the enzyme urea cycle, which is found primarily in the liver but is commonly expressed at low levels in other tissues, including the liver, brain, eyes, submandibular gland, thymus, lungs, mammary gland, heart, spleen, adrenal gland, pancreas, kidneys, intestines, bladder, prostate, testes, epididymis, skeletal muscle, vessels, erythrocytes, and skin [30]. The present results showed that the complete replacement of soybean meal with a miscellaneous meal resulted in a decrease in serum urea levels in growing pigs, which is consistent with a previous study [31]. Furthermore, triglyceride mainly participates in body fat and energy metabolism, and infection and inflammation can lead to multiple alterations in lipid and lipoprotein metabolism [32]. According to our findings, substituting soybean meal with miscellaneous meal entirely led to higher serum TG levels in growing pigs, aligning with previously reported outcomes that replacing soybean meal with 33% sunflower seed meal in the diet of broiler chickens resulted in reduced serum TG levels [14]. 

The inter-tissue transportation of amino acids is predominantly mediated by serum free amino acids. The concentration of these amino acids in the serum serves as an indicator of dietary composition and ingestion patterns [33]. Our current study shows that substituting soybean meal with miscellaneous meals did not significantly affect the levels of serum free amino acids in growing pigs. Previous experiments have demonstrated that supplementing the diets of growing pigs (33.6 ± 0.65 kg) with free amino acids resulted in elevated serum concentrations of essential amino acids such as Arg, His, Lys, Phe, Thr, Trp, and Val. This supplementation also led to increased concentrations of non-essential amino acids, including Asp, Tyr, Glu, Gly, and Ser [34]. Furthermore, pigs of similar initial body weights (22 kg and 21.8 kg) fed a low-protein diet supplemented with free amino acids exhibited higher serum concentrations of Lys and Thr than those fed a high-protein diet. Conversely, pigs in the high-protein group had higher serum concentrations of Arg, His, Ile, Leu, Met, Phe, and Val [35]. Based on these two findings, we believe that the constant protein level in the diets and the absence of exogenous free amino acid addition may explain the lack of significant differences in serum free amino acid contents in our study. 

Intestinal flora and their metabolites play an important role in animal health, and diet is one of the main factors influencing the composition of intestinal flora. It is known from previous studies that *Firmicutes* and *Bacteroidetes* are the two most abundant bacterial phyla in this breed of pigs [36]. Notably, a diet supplemented with 20% rapeseed meal considerably reduced the relative abundance of the *Bacteroidetes* phylum while tending to increase the relative abundance of the *Firmicutes* phylum [24]. Consistently, our results showed that *Firmicutes* and *Bacteroidetes* were the most dominant phyla in 50–75 kg growing pigs’ feces. In our study, *Proteobacteria* emerged as the third most abundant phylum across all three treatment groups. Previous research by Niu et al. [37] established a positive correlation between *Proteobacteria* and both crude fiber digestibility and acid detergent fiber digestibility in pigs aged 28 to 150 days. Conversely, they found a negative correlation between *Euryachaeota* and the digestibility of these same fibers within the same age group [37]. Our *t*-test analysis revealed a significant reduction in the abundance of *Euryachaeota* at the phylum level in the CM group when compared to the CON group. This suggests that a mixed meal diet can potentially improve the intestinal crude fiber-digesting flora contents in growing pigs weighing 50–75 kg. Furthermore, the analysis of microbial alpha diversity revealed that the utilization of miscellaneous meals as a complete replacement for soybean meal did not have any impact on either the Chao1 or Simpson index; these findings are consistent with Gu et al. [14]. Other studies also showed similar results for Shannon index, observed OTUs, Chao1, and whole-tree phylogenetic diversity in the colons of growing pigs (25 ± 2 kg) using a 100% replacement of rapeseed meal for soybean meal [38]. Although the increased bacterial load from the small intestine to the large intestine resulted in a difference in intestinal location from that expected, the diversity was similar between dietary groups at each location. It is important to note that research has shown that feeding fermented soybean meal to growing pigs (17.46 ± 1.97 kg) instead of regular soybean meal greatly decreased the amount of *Escherichia coli* in the colon while dramatically increasing the amount of *Lactobacillus* [39]. In a similar vein, it was also found that the addition of fermented soybean meal instead of soybean meal could significantly increase the number of lactic acid bacteria in the feces of piglets (7 kg), while concurrently reducing the total count of *Coliforms* and *Clostridium perfringens* [40]. This outcome is attributable to the fermentation of rapeseed meal. However, there is currently little research on the intestinal flora of developing pigs fed cottonseed meal and sunflower meal rather than soybean meal, necessitating more investigation. In summary, our experimental results showed no significant difference in bacterial flora, which suggests that the replacement of soybean meal with miscellaneous mixed meals (rapeseed meal, cottonseed meal, and sunflower seed meal) did not affect the colonic intestinal environment of growing pigs.

## 5. Conclusions

In conclusion, this study indicated that miscellaneous meal as a substitute for soybean meal in the diet had no significant negative impact on the growth performance, apparent nutrient digestibility, or serum free amino acid concentrations in 50–75 kg growing pigs. Furthermore, it did not alter the diversity of the fecal microbiota. Therefore, these findings suggest that miscellaneous meals (rapeseed meal, cottonseed meal, and sunflower seed meal) can serve as partial or complete substitutes for soybean meal in the diets of 50 to 75 kg growing pigs, potentially offering excellent alternative protein sources.

## Figures and Tables

**Figure 1 animals-13-03499-f001:**
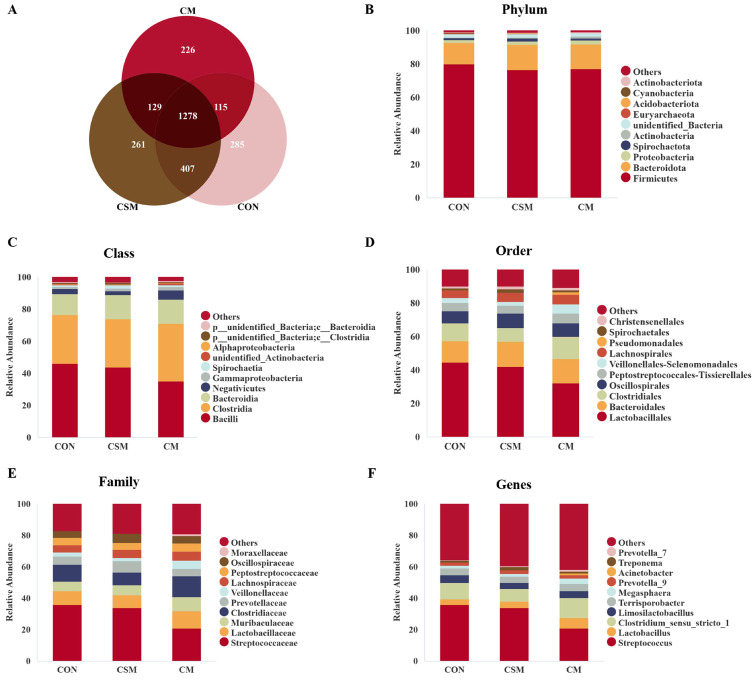
The Venn diagram and relative abundances of bacteria at the phylum, class, order, family, and genus levels. (**A**) Venn diagram. (**B**) Top 10 bacteria at the phylum level. (**C**) Top 10 bacteria at the class level. (**D**) Top 10 bacteria at the order level. (**E**) Top 10 bacteria at the family level. (**F**) Top 10 bacteria at the genus level. Abbreviations: CON, control; CSM, corn–soybean–miscellaneous meal; CM, corn–miscellaneous meal.

**Figure 2 animals-13-03499-f002:**
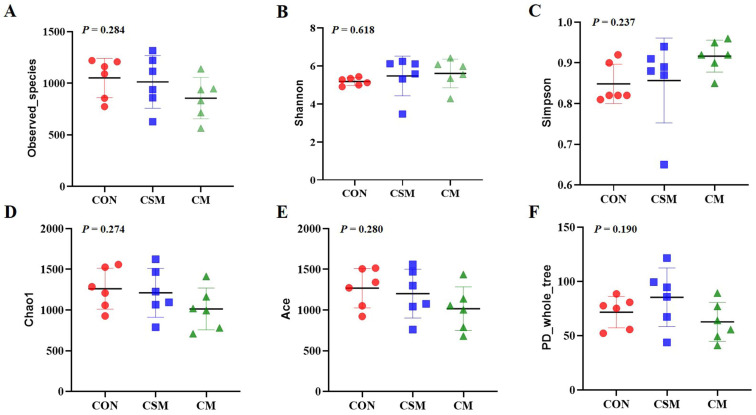
The effect of miscellaneous meal (rapeseed meal, cottonseed meal, and sunflower seed meal) as a replacement for soybean meal in the diet on the alpha diversity of colonic microbiota in 50–75 kg growing pigs. (**A**) Observed_species index. (**B**) Shannon index. (**C**) Simpson index. (**D**) Chao1 index. (**E**) Ace index. (**F**) PD_whole_tree index. Abbreviations: CON, control; CSM, corn–soybean–miscellaneous meal; CM, corn–miscellaneous meal.

**Figure 3 animals-13-03499-f003:**
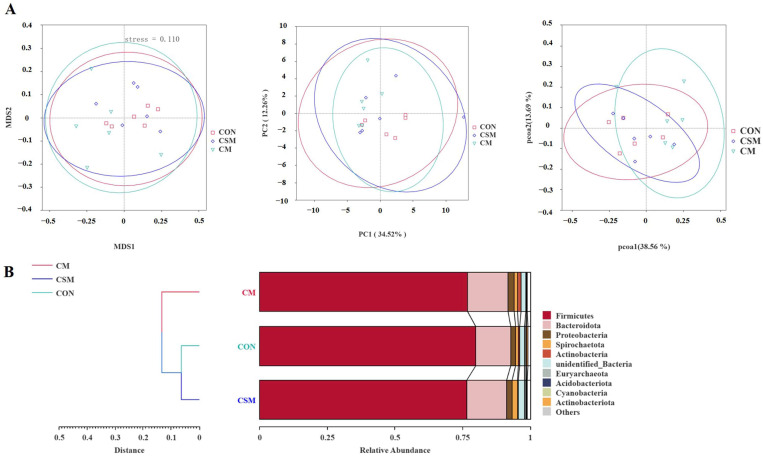
The effect of miscellaneous meal (rapeseed meal, cottonseed meal, and sunflower seed meal) as a replacement for soybean meal in the diet on the beta diversity of colonic microbiota in 50–75 kg growing pigs. (**A**) The non-metric multidimensional scaling (NMDS), principal component analysis (PCA), and principal coordinate analysis (PCoA) plots. (**B**) The unweighted pair-group method with arithmetic means (UPGMA) plot. (**C**) *t*-test plot. (**D**) The LEfSe analysis (LDA score > 3). Abbreviations: CON, control; CSM, corn–soybean–miscellaneous meal; CM, corn–miscellaneous meal. * Significant differences were derived by *t*-test for the two treatment groups (*p* < 0.05); ** Highly significant differences were derived by *t*-test for the two treatment groups (*p* < 0.01).

**Table 1 animals-13-03499-t001:** The composition and nutrient levels of the different diets for 50 to 75 kg growing pigs (air-dried basal).

Ingredients, %	CON	CSM	CM	Calculated Nutrient Levels	CON	CSM	CM
Corn	75.38	72.37	69.72	Digestive energy (Kcal/kg)	3427	3410	3394
Soybean meal	20.99	10.00		Metabolic energy (Kcal/kg)	3320	3303	3289
Rapeseed meal		4.04	7.69	Net energy (Kcal/kg)	2508	2508	2508
Cottonseed meal		4.04	7.69	Crude Protein *, %	15.3	15.0	14.7
Sunflower seed meal		4.04	7.68	Ca *, %	0.62	0.59	0.57
Soybean oil	0.42	2.15	3.67	STTD P *, %	0.25	0.27	0.29
Limestone	0.92	0.89	0.87	SID Lysine, %	0.85	0.85	0.85
CaHPO_4_	0.25	0.23	0.21	SID Met, %	0.24	0.24	0.24
NaCl	0.40	0.40	0.40	SID Thr, %	0.52	0.52	0.52
L-lysine sulfate	0.26	0.41	0.56	SID Trp, %	0.15	0.15	0.15
DL-Met	0.02	0.01		SID Val, %	0.58	0.56	0.55
L-Thr	0.05	0.09	0.12				
L-Trp		0.02	0.03				
L-Val			0.01				
L-Ile			0.04				
TiO_2_	0.4	0.4	0.4				
Vitamin–mineral premix ^(1)^	1.27	1.27	1.27				
Total	100	100	100				

^(1)^ The premix provided the following per kg of feed: VA 4 500 IU, VD2 100 IU, VE 22.5 mg, VK 3.75 mg, VB1 2.25 mg, VB2 7.5 mg, nicotinic acid 30 mg, D-pantothenic acid 11.25 mg, folic acid 0.75 mg, VB6 3 mg, VB12 0.03 mg, biotin 0.08 mg, Fe(FeSO_4_·H_2_O) 112.5 mg, Cu(CuSO_4_·5H_2_O) 6 mg, Mn(MnSO_4_·H_2_O) 4.5 mg, Zn(ZnSO_4_·H_2_O) 60 mg, I(CaI_2_O_6_) 0.14 mg, Se(Na_2_SeO_3_) 0.3 mg. * Indicates that the nutrient level is the actual value detected. Abbreviations: CON, corn–soybean meal; CSM, corn–soybean–miscellaneous meal; CM, corn–miscellaneous meal; STTD, standardized total tract digestible.

**Table 2 animals-13-03499-t002:** The nutrient information for rapeseed meal, cottonseed meal, and sunflower seed meal.

Items	Crude Protein, %	Ether Extract, %	Crude Fiber, %	Ca, %	STTD P, %
Rapeseed meal	38.6	1.4	11.8	0.65	0.25
Cottonseed meal	47.0	0.5	10.2	0.25	0.28
Sunflower seed meal	36.5	1.0	10.5	0.27	0.29

**Table 3 animals-13-03499-t003:** Effect of miscellaneous meal (rapeseed meal, cottonseed meal, and sunflower seed meal) as a replacement for soybean meal in the diet on growth performance of 50–75 kg growing pigs ^1^.

Items	CON	CSM	CM	SEM	*p*-Value ^(1)^
IBW (kg)	49.99	50.30	51.64	0.493	0.368
FBW (kg)	75.76	76.99	78.13	0.645	0.342
ADG (kg/d)	1.10	1.14	1.17	0.024	0.450
ADFI (kg/d)	2.57	2.64	2.66	0.032	0.478
F/G	2.35	2.32	2.29	0.033	0.827

^1^ Values are means and standard error of the means (n = 6). Abbreviations: IBW, initial body weight; FBW, final body weight; ADG, average daily gain; ADFI, average daily feed intake; F/G, feed gain ratio; CON, corn–soybean meal; CSM, corn–soybean– miscellaneous meal; CM, corn–miscellaneous meal. ^(1)^ The overall *p*-values were obtained by one-way ANOVA for the three treatment groups.

**Table 4 animals-13-03499-t004:** The results of analysis of covariance for the effect of initial basic body weight on final body weight and average daily gain.

Items	Degrees of Freedom	FBW				ADG			
		Sum of Squares	Mean Square	F	*p* ^(1)^	Sum of Squares	Mean Square	F	*p* ^(1)^
IBW	1	0.008	0.008	0.001	0.975	0.0005	0.0005	0.043	0.839

Abbreviations: IBW, initial body weight; FBW, final body weight; ADG, average daily gain. ^(1)^ The overall *p*-values were obtained by analysis of covariance (ANCOVA) for the three treatment groups.

**Table 5 animals-13-03499-t005:** Effect of the replacement of soybean meal with miscellaneous meal (rapeseed meal, cottonseed meal, and sunflower seed meal) on the growth performance of 50–75 kg growing pigs of different sexes.

Items	CON		CSM		CM		*p*-Value ^(1)^		
	Barrows	Gilts	Barrows	Gilts	Barrows	Gilts	Treatment	Sex	Treatment × Sex
FBW	76.76	74.76	76.84	77.13	77.78	78.49	0.561	0.854	0.804
ADG	1.10	1.07	1.13	1.15	1.16	1.19	0.236	0.934	0.782

Abbreviations: FBW, final body weight; ADG, average daily gain; CON, corn–soybean meal; CSM, corn–soybean– miscellaneous meal; CM, corn–miscellaneous meal. ^(1)^ The overall *p*-values were obtained by analysis of two-way ANOVA for the three treatment groups.

**Table 6 animals-13-03499-t006:** Effect of miscellaneous meal (rapeseed meal, cottonseed meal, and sunflower seed meal) as a replacement for soybean meal in the diet on the apparent nutrient digestibility of 50–75 kg growing pigs ^1^.

Items	CON	CSM	CM	SEM	*p*-Value ^(1)^
Crude Protein, %	73.27	71.83	70.73	0.916	0.553
Crude Fat, %	79.89	80.26	80.47	0.429	0.854
Gross Energy, %	82.82	82.50	80.80	0.428	0.113

^1^ Values are means and standard error of the means (n = 6). ^(1)^ The overall *p*-values were obtained by one-way ANOVA analysis for the three treatment groups. Abbreviations: CON, control; CSM, corn–soybean–miscellaneous meal; CM, corn–miscellaneous meal.

**Table 7 animals-13-03499-t007:** Effect of miscellaneous meal (rapeseed meal, cottonseed meal, and sunflower seed meal) as a replacement for soybean meal in the diet on serum biochemical parameters of 50–75 kg growing pigs ^1^.

Items	CON	CSM	CM	SEM	*p*-Value ^(1)^
TP (g/L)	63.50	63.61	65.44	0.613	0.369
CRE (umol/L)	119.97	124.62	133.29	3.284	0.256
AST (U/L)	44.25	47.11	42.24	2.660	0.776
ALP (U/L)	164.17	155.19	155.95	6.557	0.843
ALT (U/L)	45.40 ^b^	48.16 ^ab^	59.07 ^a^	2.431	0.041 *
ALB (g/L)	3.65	3.82	4.13	0.119	0.258
UREA (mmol/L)	5.12 ^a^	4.15 ^ab^	3.45 ^b^	0.255	0.017 *
GLU (mmol/L)	3.35	4.15	4.18	0.180	0.098
TG (mmol/L)	0.42 ^b^	0.59 ^b^	1.51 ^a^	0.166	0.007 *
CHO (mmol/L)	0.82	0.79	0.81	0.022	0.906
HDL-C (mmol/L)	0.86	0.89	0.91	0.026	0.699
LDL-C (mmol/L)	1.57	1.57	1.76	0.059	0.330

^1^ Values are means and standard error of the means (*n* = 6). ^(1)^ The overall *p*-values were obtained by one-way ANOVA analysis for the three treatment groups. Abbreviations: TP, total protein; CRE, creatinine; AST, aspartate aminotransferase; ALP, alkaline phosphatase; ALT, alanine aminotransferase; ALB, albumin; UREA, urea; GLU, glucose; TG, triglyceride; CHO, cholesterol; HDL-C, high-density lipoprotein cholesterol; LDL-C, low-density lipoprotein cholesterol; CON, control; CSM, corn–soybean–miscellaneous meal; CM, corn–miscellaneous meal. ^ab^ Values in the same row with different superscripts differ significantly (*p* < 0.05). * Significant differences were derived by one-way ANOVA for the three treatment groups (*p* < 0.05).

**Table 8 animals-13-03499-t008:** Effect of miscellaneous meal (rapeseed meal, cottonseed meal, and sunflower seed meal) as a replacement for soybean meal in the diet on serum free amino acid contents of 50–75 kg growing pigs ^1^.

Item	CON	CSM	CM	SEM	*p*-Value ^(1)^
Essential amino acids					
Lys	449.01	533.19	559.66	53.660	0.891
Met	65.99	64.34	68.42	2.660	0.922
Thr	212.19	239.19	229.82	20.031	0.870
Val	350.01	397.17	387.51	23.412	0.711
Ile	138.34	168.09	159.01	10.145	0.499
Leu	274.09	323.87	305.44	18.662	0.574
Phe	117.22	130.07	131.90	6.809	0.659
Trp	70.24	83.69	65.47	4.704	0.257
His	75.72	72.39	70.79	3.714	0.873
Arg	259.71	335.07	302.91	23.565	0.450
Non-essential amino acids					
Ser	215.86	197.90	213.12	12.622	0.840
Ala	876.05	862.56	972.06	74.304	0.825
Gly	1600.36	1450.84	1530.49	62.899	0.652
Tyr	109.11	115.93	107.62	6.237	0.861
Glu	325.18	313.77	462.57	32.034	0.102
Asp	27.86	26.87	36.35	2.315	0.190
Cys	31.48	28.31	24.17	2.331	0.477
Hypro	67.33	75.33	77.60	4.155	0.600
Pro	393.60	364.00	411.20	23.478	0.728
Non-protein amino acids and derivatives/metabolites of amino acids					
Tau	115.19	110.43	118.17	6.260	0.891
NH3	263.26	295.56	348.38	20.494	0.242
Cit	84.05	85.14	85.63	4.908	0.992
Orn	110.09	110.57	120.05	7.464	0.845
∑EAA	1942.30	2263.38	2215.46	150.325	0.670
∑NEAA	3575.99	3435.53	3753.72	201.260	0.829

^1^ Values are means and standard error of the means (n = 6). ^(1)^ The overall *p*-values were obtained by one-way ANOVA analysis for the three treatment groups. Abbreviations: EAA, essential amino acids; NEAA, non-essential amino acids; Lys, lysine; Met, methionine; Thr, threonine; Val, valine; Ile, isoleucine; Leu, leucine; Phe, phenylalanine; Trp, tryptophan; His, histidine; Arg, arginine; Ser, serine; Ala, alanine; Gly, glycine; Tyr, tyrosine; Glu, glutamicacid; Asp, asparticacid; Cys, cystine; Hypro, hydroxyproline; Pro, proline; Tau, taurine; Cit, citrulline; Orn, ornithine.

## Data Availability

The original contributions presented in the study are included in the article, and further inquiries can be directed to the corresponding author.

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
