# Peer review of "Effect of Miscellaneous Meal Replacements for Soybean Meal on Growth Performance, Serum Biochemical Parameters, and Gut Microbiota of 50–75 kg Growing Pigs"

_animals, 2023, doi:10.3390/ani13223499_

Round 1
Reviewer 1 Report
Comments and Suggestions for Authors
The studies concerning the alternatives of regular feedstuff particular soybean meal are very essential and meaningful under the currently high cost period. The authors conducted the study to investigate the effects of miscellaneous meal replace soybean meal on growth performance, nutrient apparent digestibility, serum biochemical parameters, serum-free amino acid contents, and gut microbiota of growing pigs with 50-75 kg. This study demonstrated that the miscellaneous meal contains rapeseed meals, cottonseed meals, and sunflower seed meals as the alternative of soybean meal in the diet had no significant negative effects on growth and nutrients digestibility of growing pigs, which could give very useful information to feed industry and swine nutrition. The experimental design and whole written were fine. However, there are some issues need to be solved by the authors.
1. The authors should provide the analyzed nutrient information of rapeseed meals, cottonseed meals, and sunflower seed meals.
2. Why the authors set the replace ratio of soybean meal as 10.99% in the CSM diet?
3. Some nutrients levels in the formula should be measured, such as CP, Ca, P.
4. Actually, there is visible difference in initial BW among the treatments (49.99 vs 51.64) that might influence the whole performance, the authors should explain the reason why did not select the pigs with similar BW at the beginning.
5. In the discussion section, the authors should build up the logical relationship between each parameter. For instance, whether the effects of miscellaneous meal replace soybean meal on growth performance is attribute to the effect on nutrient digestibility. Apparently, the dietary effects on five parameters in the discussion section were discussed separately.
6. There are many formative and grammatical issues in the text that should be carefully modified by the authors.
Comments on the Quality of English Language
There are many formative and grammatical issues in the text that should be carefully modified by the authors.
Reviewer 2 Report
Comments and Suggestions for Authors
Summary of the work:
This summary looked at replacing soybean as a plant protein source for growing pigs from 50 – 75 kg. A total of three diets that looked at different combinations of plant protein sources was assessed. Pigs were on study for 24 days, in replicates of 6. Pigs were kept in a controlled temperature environment and were fed a total of three times a day. Pigs were weighed at the end of the study, a blood sample was collected, and feces was collected over the last three days of the study. There were no differences in the growth of the pigs, or the digestibility of the diets. There were no significant effects on the blood serum chemistry as well. There were some minor differences in the microbiome of the pigs across treatments including differences of Euryachaetos at the phylum level and Desulfobacterota at the genus level. The authors conclude that if these alternatives are fed from 50-75 kg of the growing period, there are not significant differences.
Comments:
Last line in first introduction paragraph – words missing from sentence and it is not grammatically correct. For example, the sentence ends in a comma and not a period.
“Alternative native protein feedstocks” – what does this mean?
“Protein raw materials” This needs to be specific and state you are looking at grains. Remember that pigs can receive protein from other sources such as whey and blood meal as a weaning.
“fattening period” is slang. Be specific, what is pig weight range in this time frame.
“miscellaneous meals” does not truly specify that you are focusing on grain sources of protein. Again there is also blood meal, feather meal, bone meal.
Pigs – were these barrows, gilts, boars? Did pens have both barrow/boar and gilts in them, or just one sex?
Traditionally, growing pigs are fed ad libitum. Why are these pigs fed three times a day? Typically if a growing pig is not being fed ad libitum, it is in a metabolism study. However, metabolism studies are not representative of the true growth of pigs as the amount of feed they can eat is limited.
I am confused. If pigs had ad libitum access to feed, why are they fed three times a day? Why is there not a feeder that is filled with feed constantly in front of the pig? This would be how a growing pig is fed on a commercial farm.
While the diets are balanced across metabolic and net energy, there is a great difference in the amount of fat that is included in the diets. This can attribute to feed intake, particularly in a situation when the pigs are not in a temperature controlled environment. – please comment on this.
You list that you only weighed pigs at the end of the study, however; you list a starting body weight. Please correct how often the pigs were weighed or list how you got the initial body weight.
How was the serum stored between the time of collection and analysis?
Was the feed intake measured daily? Between each feeding?
The replicate is not the experimental unit. Pen is the experimental unit. Did you include any fix effects, such sex, in the model? Growing pigs do have differences in growth based on sex. Therefore, this needs to be in the model if there was more than one sex of pig used in the study (sex of the pigs needs to be added as previously mentioned).
In the gut microbiota results, you mentioned that you ran T tests, this information needs to be added to the statistical analysis portion of the manuscript.
In recent years, the price of soybean meal has become very cheap in the United States, particularly in the Midwest region. Your discussion needs to take into consideration geographical locations and current economies on which plant proteins are being formulated into the swine diet. This will differ based on the economy and location in which the pigs are located. Also take into consideration locations in which these alternatives will not grow well, and thus are actually more expensive than soybean meal because they would need to be transported across great distances to be brought to the feed mill.
Why are you discussing sows in this paper? This sentence is out of place, sows and growing pigs are not comparable as they are at different physiological states.
When discussing the serum biochemistry, you did a nice job of discussing what the raised/lowered levels indicate. Discuss if there are any concerns. For example, if elevated ALT levels indicate liver damage, what does this mean for the growing pig? Should pig producers be concerned about a potential increase in vet bills or survivability?
I appreciate the comparisons of the microbiome. Could you comment more of on the function and importance of the mentioned bacteria? If you point out common phylum, what does that phylum do in the intestine? While it is nice to know what bacteria is present, I think it is equally important to comment on the significance of the presence of the bacteria, such as are the metabolites important or are the bacteria necessary for fiber digestion? You briefly comment on some of this, but I think it needs to be expanded.
I would appreciate some discussion on the following: Why were the pigs not raised similarly to commercial farms? Why was a metabolism study not conducted for this as it appears that was the ultimate goal of the paper? Why did you only have the pigs on study for such a short period? What would happen if the pig was on these diets from 50 kg till market? Potential sex differences? Would there be any impact on the carcass composition?
I think overall the discussion is a good start, but it needs to have a more details added. The manuscript as a whole provides nice information about soybean meal alternatives.
Comments on the Quality of English LanguagePlease review the introduction of the paper. There are some sentences that appear to be missing words and some grammatical errors that need to be addressed.
Round 2
Reviewer 2 Report
Comments and Suggestions for Authors
I think this paper is more well rounded. Requested information is added in and the details described in the experiment are more representative and informative. The discussion is also improved.